# Tomato Comprehensive Quality Evaluation and Irrigation Mode Optimization with Biogas Slurry Based on the Combined Evaluation Model

**Jian Zheng [1,2,3,*], Xingyun Qi [1,2,3], Cong Shi [1,2,3], Shaohong Yang [1,2,3] and You Wu [1]**

1   College of Energy and Power Engineering, Lanzhou University of Technology, Lanzhou 730050, China; slqxy_2019@163.com (X.Q.); congshi1997@126.com (C.S.); yangshaohong30@163.com (S.Y.); wuyou@lut.edu.cn (Y.W.)
2   Key Laboratory of Complementary Energy System of Biomass and Solar Energy, Lanzhou 730050, China
3   China Northwestern Collaborative Innovation Center of Low-Carbon Urbanization Technologies, Lanzhou 730050, China
*   Correspondence: zhj16822@126.com

**Abstract:** Scientific and reasonable water and fertilizer regimes positively affected crop growth, yield, fruits quality and soil environment improvement. As a liquid quick-acting organic fertilizer to substitute chemical fertilizers, biogas slurry has been widely used in agricultural production. However, the lack of research on the proper comprehensive quality evaluation model and irrigation mode under biogas slurry limits the promotion and large-scale application of biogas slurry in agricultural production. In this study, three biogas slurry (BS) ratio (1:4BS, 1:6BS, 1:8BS; volume ratio of biogas slurry to water), three irrigation levels (W1, W2, W3) and three fertilizer control treatments (CF1, CF2, CF3) were conducted in field experiments. Eleven single indexes from four type qualities (external quality, taste quality, nutrition quality, storage and transportation quality) were adopted to establish the comprehensive evaluation index system of tomato. The principal component analysis, grey correlation analysis, membership function analysis and TOPSIS analysis model (based on the combination of objective entropy method and subjective analytic hierarchy process) were used to estimate the comprehensive quality of tomato fruits. Moreover, the objective combination evaluation mode based on overall diversity was used to evaluate the results obtained from the four independent comprehensive evaluation methods. The aim is to mitigate inconsistencies of multi-attribute evaluation models. The results showed that biogas slurry application was beneficial to the accumulation of aboveground biomass under the same irrigation amount, which can effectively improve the sugar to acid ratio and lycopene content of tomato. T3 (1:4BS, W3) and T1 (1:4BS, W1) obtained the highest yield and water use efficiency (WUE), respectively. The results of Kendall consistency test and Pearson correlation coefficients showed that there were good compatibility and high consistency among the four independent comprehensive evaluation models, and the combined quality evaluation model can be performed directly. As the correlation coefficients between combined evaluation model and each of four independent methods reached 0.965, the combined evaluation model was capable of reducing the differences of four independent comprehensive evaluation model. The combined quality evaluation results showed that T2 (1:4BS, W2) recommended strongly in this study could effectively improve the yield, quality and WUE of tomato.

**Keywords:** biogas slurry; tomato; yield; comprehensive quality; overall difference combination evaluation model

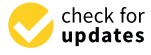



## 1. Introduction

The rapid development of facility-based agriculture has effectively alleviated the contradictions of regional restrictions and seasonal shortages in traditional agricultural vegetable production [1]. In recent years, facility-based vegetables have become the main

source of people's daily vegetable consumption. However, with the improvement of human living environment, quality of life and modernization of agricultural production, consumers are paying more and more attention to the healthy diet and vegetables quality [2]. At the same time, tomatoes (*Solanum lycopersicum* L.) not only are rich in polyphenols [3], fibre, vitamin C and numerous antioxidants with high nutritional value, but also are important sources of tomato lycopene [4,5]. Thus, better fruits quality has been a common demand of tomato producers and consumers.

The growth, yield and fruits quality of tomato are affected by many factors, including genes, climate, water and fertilizer supply, field management, fruits storage and transportation [6]. Recently, a large number of scholars have devoted increasing attention to the ways of increasing tomato production and improving fruits quality. Herein, the regulation of water and fertilizer is the simplest and the most direct influencing factor that is artificially controllable [7]. Suitable water-fertilizer integrated technology and its regulation mode can effectively promote crop growth, improve the soil environment in the root zone, increase the photosynthetic rate of plants, and enhance crop quality [8,9]. Biogas slurry is the afterproduct of anaerobic fermentation in biogas engineering, that is rich in nutrients required for plant growth (N, P, K, humic acid and plant hormones, etc.), trace elements that are beneficial to the soil environment (amino acids, organic matter, organic and inorganic salts, gibberellin, antibiotics and vitamins B, etc.), and other substances that helps plant to resist diseases, pests and adverse environment (monosaccharides, free amino acids and antibiotics, etc.) [10–13]. The agricultural application of biogas slurry not only increases crops yield and quality [14], but also improves soil environment in the farming area [15]. The characteristics of biogas slurry with high water and low fertilizer meet the requirement of water-fertilizer integrated technology. At present, the research on water and biogas slurry integrated technology mainly focuses on irrigation parameters as well as crop growth, yield and quality responses of plants [16–18]. However, the response and evaluation of the comprehensive nutritional quality of crops (external quality, taste quality, nutrition quality as well as storage and transportation quality) under water/biogas slurry integrated irrigation conditions was still unclear. Crop quality, the key factor determining the economic benefits, is influenced by coupling effects of multiple factors. Meanwhile, various consumers have different requirements for crop quality, and it is difficult to evaluate crops quality using a single index.

The agricultural production mode recommended by agricultural production evaluation models has played a positive role in guiding agricultural production and can effectively improve economic benefits. Evaluation models have now evolved from simple hierarchical analysis [19,20] and entropy weighting methods [21] into multi-attribute comprehensive evaluation, such as principal component analysis [22,23], grey correlation analysis [24], affiliation function analysis [25] and TOPSIS analysis [26–28]. However, according to the comparison of evaluation models, obvious diversity of evaluation results restricts objective decision-making and scientific field management, due to the discrepancies of the evaluators' subjective behavior, the selectivity of the method structure and the discarding of evaluated information [29]. Therefore, it is necessary to seek a multi-method integrated combined evaluation model to estimate the comprehensive nutritional quality of tomato fruits. The study shows that the overall difference combination evaluation model can overcome the inconsistent findings of multiple single evaluation methods and make the results more objective and accurate [29].

Consequently, a systematic investigation is carried out on the effects of different biogas slurry ratios and irrigation rates on facility tomato growth, yield, water use efficiency (WUE) and quality. In this study, the water/biogas slurry integrated irrigation was adopted, including three biogas slurry ratios, three irrigation levels and three chemical fertilizer control treatments. The appearance quality of tomato fruit (single fresh fruit weight, fruit shape ratio), the flavor quality (soluble sugar, titratable acid, sugar acid ratio, soluble solid), the health quality (vitamin C, soluble protein, lycopene) as well as storage and transportation quality (fruit water content, fruit hardness) were selected for the comprehensive nutritional

tomato fruits quality evaluation using the principal component analysis method, grey correlation analysis method, the membership function analysis method as well as TOPSIS analysis model based on the combined weighting of objective entropy weight method and subjective analytic hierarchy process. The combined evaluation model was introduced to generally estimate the evaluation values from each independent comprehensive evaluation model, so as to obtain the optimal water/biogas slurry integrated irrigation regime. This study takes greenhouse tomato, the most representative planting crop in the arid region of Northwest China, as the research object. Combined with the local greenhouse economic crops that are easy to promote, the integrated hole irrigation technology of water/biogas slurry is used to continuously promote and improve farmers' income and consumers' food quality. At the same time, this study aims to not only provide a theoretical basis and technical support for the promotion of integrated water/biogas irrigation technology, the reduction of chemical fertilizers and the efficient use of biogas liquid in the facility agriculture, but also put forward some new ideas for the comprehensive quality evaluation of tomato fruits.

## 2. Materials and Methods

### 2.1. Experimental Site

The experiment was conducted in a vegetable cultivation greenhouse (36°01′ N, 103°46′ E, 1835 m above sea level) with water-fertilizer integrated irrigation technology in Weiling country, Qilihe District, Lanzhou City of Gansu Province with an altitude of 1835.7 m (Figure 1). The experimental area belongs to the temperate continental climate with year-round drought and sufficient sunshine. The average annual temperature is 10.3 °C, and the temperature difference between day and night reaches 12~18 °C. The frost-free period lasts about 150 days. The average annual precipitation and evaporation are 327 mm and 1158.0 mm, respectively. The length, width and height of ridge-structure greenhouse are 50 m, 10.5 m and 4 m, respectively. The greenhouse is equipped with a portable automatic weather station which continuously monitor meteorological data.

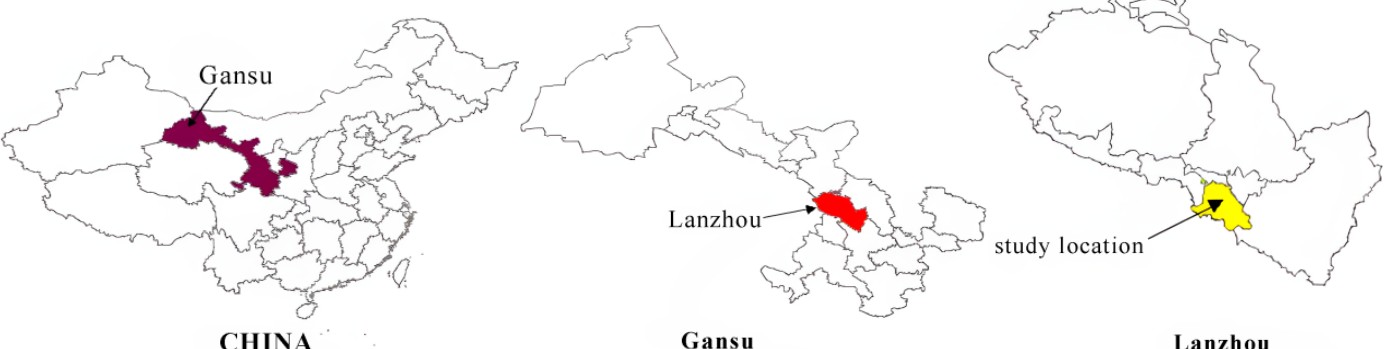

**Figure 1.** The location of experiment site.

### 2.2. Experimental Materials

The tomato variety (*Solanum lycopersicum* L.) used in the experiment was 'Hongbao 5'. The biogas slurry took cattle manure as raw material collected from the Holstein Dairy Cattle Breeding Center of Lanzhou city. Cattle manure was fermented at 37 °C under anaerobic condition for 2 months, and then was filtered with four layers of gauze (32 mesh), sterilized and applied when physical and chemical properties became constant. The main characteristics of the biogas slurry were: total nitrogen (N) of 1.038 g $L^{-1}$, total phosphorus (P) of 0.553 g $L^{-1}$, total potassium (K) of 1.201 g $L^{-1}$, organic matter content of 10.65 g $L^{-1}$, pH of 7.89, the conductivity of 23.59 ds $m^{-1}$, and the viscosity of 1.869 ds $m^{-1}$. It should be noted that the biogas slurry was placed in a plastic bucket and left open to settle for two months before the experiment, when used the upper clear liquid was taken and the larger suspended particles were filtered out using 4 layers gauze of 32 mesh. Meanwhile, the

relevant physical and chemical properties of the biogas slurry were measured every seven days during the test to ensure the basic stability of the physical and chemical properties of the biogas slurry used during the test period.

According to the international soil texture classification standard, soil texture of experimental fields was loam clay with clay of 54.6%, silt of 33.8% and sand of 11.60%. The average soil capacity above the 1 m soil layer of cultivated area was 1.37 g cm$^{-3}$. The maximum water holding capacity was 23.5% (weight water fraction). The average nutrient content of the 0-60 cm soil layer before transplanting were organic matter of 9.3 g kg$^{-1}$, total N of 0.683 g kg$^{-1}$, total P of 1.275 g kg$^{-1}$ and total K of 1.586 g kg$^{-1}$. Soil pH was 7.92 before the experiment.

### 2.3. Experimental Design

In this case, biogas slurry concentrations, i.e., 1:4, 1:6 and 1:8 (biogas slurry: water, volumetric ratio), and three irrigation levels, i.e., W1, W2, W3 were set in the orthogonal experiments. Three chemical fertilizer controls, i.e., CF1, CF2 and CF3 were irrigated with 0.6, 0.8 and 1.0 W, respectively. Detailed information of total 12 treatments were shown in Table 1. The irrigation amount W were calculated by formula:

$$W = Kc \cdot A \cdot Ep$$

where Kc is the crop-evaporator dish coefficient, taken as 0.6 (W1), 0.8 (W2) and 1.0 (W3); A is the area of per plant (30 cm × 50 cm); Ep is the evaporation amount of the standard evaporating dish (Φ20 cm) during irrigation interval. The location of evaporating dish was adjusted for keeping consistent with the canopy height. The irrigation frequency was 1 time per 2 days. The fertilization regime under chemical fertilizer control treatments was identical to local farmers with 78 kg ha$^{-1}$ urea (46.4% N), 94.5 kg ha$^{-1}$ diammonium phosphate (40% P$_2$O$_5$) and 97.5 kg ha$^{-1}$ potassium sulphate (45% K) for four times topdressing during the whole growth stage.

**Table 1.** Experimental scheme for irrigation.

| Treatments | Fertilizer or Digestate Application | Irrigation Level |
|---|---|---|
| CF1 |  | W1 |
| CF2 | Chemical fertilizer | W2 |
| CF3 |  | W3 |
| T1 |  | W1 |
| T2 | biogas slurry: water, 1:4 | W2 |
| T3 |  | W3 |
| T4 |  | W1 |
| T5 | biogas slurry: water, 1:6 | W2 |
| T6 |  | W3 |
| T7 |  | W1 |
| T8 | biogas slurry: water, 1:8 | W2 |
| T9 |  | W3 |

The experiment started on 7 April 2019 and finished on 31 July 2019. Tomato plants were transplanted at the three or four true-leaf stage. After transplanting, 2000 mL water was applied for each plant to better survival. The single-ridge mulching technology used widely by the local farmers was adopted with the ridge height of 20 cm and the ridge distance of 50 cm. Ten plants were planted in each ridge with the plant spacing of 30 cm. Each treatment was replicated three times and arranged randomly. In order to eliminate the infiltration of water and fertilizer between the experimental plots, the geotextiles were arranged between two adjacent plots in 1 m soil depth. Hole irrigation technology was combined with water and biogas slurry integrated technology in this experiment. The hole distance was 5 cm away from the plant roots along the ridge, the hole diameter and depth

were 5 cm and 7 cm, respectively (Figure 2). The whole growing period was divided into seedling stage (7 April–30 April), flowering stage (1 May–19 May), and fruit enlargement stage (20 May–10 June) and maturity stage (11 June–31 July).

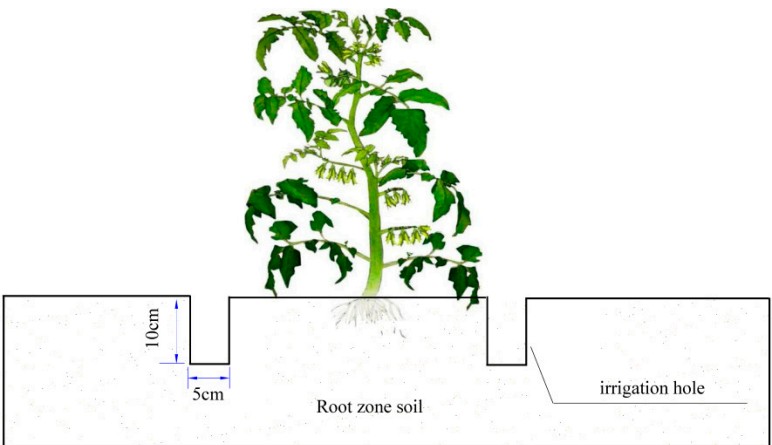

**Figure 2.** Schematic diagram of irrigation holes.

*2.4. Measurement and Evaluation Methods*

2.4.1. Measurement of Tomato Growth

Random three samples under each treatment were taken at the end of each growing stage. The roots, stems, leaves and fruits were separately weighed for fresh matter. Aboveground and root dry weight were dried using oven at 105 °C for 2 h, and then at a constant temperature of 75 °C. The dry matter of roots, stems, leaves and fruits were measured by an electronic scale with a precision of 0.01 g [18].

$$\text{Root to shoot ratio = root dry matter/aboveground dry matter}$$

The leaf area of tomato plant was determined by image method. Leaf area index (LAI) is the leaf area per unit area [30]. The length of main root was measured using a ruler (the precision of 0.01 cm).

2.4.2. Measurement of Tomato Growth

Two or three tomato fruits with similar ripeness, size and color were selected from each treatment when the third spikes of tomato fruits were ripe. After the determination of the appearance quality and hardness of fruits, the pulp was used to measure the intrinsic nutritional quality of tomato fruits. Soluble solids were determined by a WAY-2S Abbe refractometer [22]. Vitamin C was measured by Molybdenum blue colorimetric method [23]. Total soluble sugar was tested using anthrone method [24]. Organic acid was determined by acid-base titration method [25]. Soluble protein was determined by Coomassie Brilliant Blue G-250 staining method [26]. Lycopene was measured by UV spectrophotometer. Fruit firmness was measured using a HP-30 hardness tester [2]. Fruit moisture content was determined after drying. The single fruit weight was measured with an electronic scale (the precision of 0.01 g). The fruit shape ratio was obtained by measuring the transverse and longitudinal diameters with an electronic vernier caliper with the precision of 0.02 mm (fruit shape ratio = fruit longitudinal diameter/fruit transverse diameter).

2.4.3. Tomato Yield and Water Use Efficiency

Three plants were randomly selected from each treatment to calculate tomato yield. The cumulative weight from the first to the fifth spikes of fruits was determined as the final yield.

$$\text{WUE} = Y_a/I$$

where WUE is the water use efficiency, kg·m$^{-3}$; Y$_a$ is the yield per plant, kg; I is the amount of irrigation, m$^3$.

### 2.4.4. Evaluation Method

(1)    Kendall consistency test

The Kendall consistency coefficient test mainly conducts the consistency evaluation of evaluation objects ($n$) under evaluation methods ($m$), and analyzes the conformity degree of the sample data. The consistency coefficients were obtained according to the following formulas:

$$T = \frac{12 \sum\limits_{i=1}^{m} R_i^2}{m^2 n(n^2 - 1)} - \frac{3(n+1)}{n-1} \tag{1}$$

$$\chi^2 = m(n-1)T \tag{2}$$

$$R_i = \sum_{i=1}^{m} y_{ij} \tag{3}$$

where $T$ is the consistency test coefficient; $\chi^2$ is the test statistic, which follows a $\chi^2$ distribution with $(n-1)$ freedom degree, and when $\chi^2 \geq \chi_{\frac{\alpha}{2}}^2 (n-1)$, each independent evaluation method satisfies the consistency; $y_{ij}$ is the ranking value of the $i$ evaluation object under the $j$ evaluation method.

(2)    overall difference combination evaluation

Four independent evaluation methods were adopted to evaluate the comprehensive quality of tomato fruits firstly, namely principal component analysis [22], grey correlation analysis [24], membership function analysis [25] and TOPSIS analysis model (based on the combination of objective entropy method and subjective analytic hierarchy process) [26]. The Kendall consistency test was used to examine the compatibility of the results from each evaluation method [31]. Lastly, the overall difference combination evaluation [29] was used to comprehensively analyze the evaluation values of each independent evaluation model. The specific process was:

The matrix W was constructed according to principal component analysis method, grey relational degree analysis method, membership function analysis method and TOPSIS analysis model. Without losing generality, the values obtained by $m$ ($m \geq 3$) evaluation methods for $n$ ($n \geq 3$) evaluation objects was expressed as:

$$W = \left[ w_{ij} \right]_{n \times m} = \begin{bmatrix} w_{11} & \cdots & w_{1m} \\ \vdots & \ddots & \vdots \\ w_{n1} & \cdots & w_{nm} \end{bmatrix} \tag{4}$$

To ensure the comparability between the assessed values, Equation (4) was normalized to obtain the standardized matrix $W^*$:

$$W = \left[ w_{ij}^* \right]_{n \times m} = \begin{bmatrix} w_{11}^* & \cdots & w_{1m}^* \\ \vdots & \ddots & \vdots \\ w_{n1}^* & \cdots & w_{nm}^* \end{bmatrix} \tag{5}$$

where, $w_{ij}^* = \frac{w_{ij} - \overline{w_j}}{s_j}, (i = 1, 2, \cdots, n; j = 1, 2, \cdots, m); s_j = \sqrt{\frac{1}{n-1} \sum\limits_{i=1}^{n} (w_{ij} - \overline{w_j})^2}$, $w_{ij}$ is the evaluation value of the $(i = 1, 2, \cdots, n)$ evaluation object in the $j = (1, 2, \cdots, m)$ evaluation method.

The real symmetric matrix $H$ based on the matrix $W^*$ was obtained, showing $H = (W^*)^T W^*$. The maximum eigenvalue of $H$ and its corresponding standard eigenvector $\lambda^*$ were calculated. The combined weight vector $\lambda$ was determined based on $\lambda^*$,

showing $\lambda_i = \frac{\lambda_i^*}{\sum\limits_{i=1}^{m} \lambda_i^*}$, and then the combined evaluation vector value $Z = (Z_1, Z_2, \cdots, Z_M)$,
$z_i = \lambda_1 w_{i1}^* + \lambda_2 w_{i2}^* + \cdots + \lambda_m w_{im}^*, i = 1, 2, \cdots, n$ was determined. Finally, the evaluation value of tomato fruit quality indices was sorted.

### 2.5. Tomato Comprehensive Quality Evaluation Index System

Tomato fruit quality is comprehensively affected by multiple factors. In order to acquire the optimal fruit quality under various treatments, eleven quality indicators of the external quality (single fresh fruit weight, fruit shape ratio), the taste quality (soluble sugar, titratable acid, sugar-acid ratio, soluble solids), the nutrition quality (vitamin C, soluble protein, lycopene), and the storage-transportation quality (fruit moisture content, fruit firmness) were structured as a comprehensive tomato quality evaluation system, as shown in Figure 3.

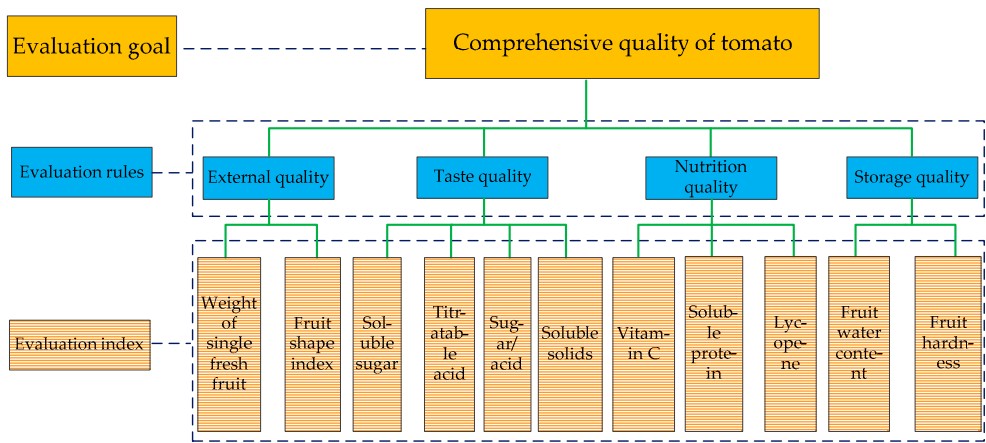

**Figure 3.** Evaluation system of comprehensive tomato fruit quality.

### 2.6. Statistical Analysis

The entropy weight method weight, grey correlation analysis, membership function analysis method, TOPSIS method and Kendall consistency test were performed by Excel 2019 (Office2019. Microsoft Corp, Redmond, WA, USA). Principal component analysis, variance analysis and Pearson correlation analysis were performed by SPSS24.0 (Origin Lab Corporation, Northampton, MA, USA). MATLAB6.5 software (Math Works was used to perform the overall difference combination evaluation model. One–way ANOVA and Duncan's multiple comparisons were used to determine significant differences between the treatments.

## 3. Results

### 3.1. Tomato Leaf Area and Dry Matter

Reasonable irrigation levels and biogas slurry application can effectively promote the formation tomato leaf area, above-ground dry matter, root dry weight, main root length and root shoot ratio (Table 2). The interaction between biogas slurry application and irrigation levels affected tomato leaf area significantly ($p = 0.351$), however, there was no significant effect on other growth indicators. Tomato leaves are important organs for photosynthesis. Appropriate leaf area plays a major role in the regulation of tomato photosynthetic substances production and transpiration. Among treatments, the leaf area under T3 reached the maximum value (9304.99 cm$^2$), which was 19.23%, 4.5% and 9.1% higher than those of CF3, T6 and T9, respectively. Conversely, the CF1 obtained the smallest leaf area of 6512.93 cm$^2$. Under the same level of fertilization, tomato leaf area increased significantly with increasing irrigation levels, showing CF3 > CF2 > CF1, T3 > T2 > T1, T6 > T5 > T4, T9 > T8 > T7 ($p < 0.001$). With the same irrigation level, the largest leaf area was obtained under 1:4 biogas slurry, which were 8253.87 cm$^2$ under T1, 9030.43 cm$^2$ under

T2 and 9304.99 cm$^2$ under T3, respectively. The leaf area index can directly reflect the growth of tomato plants. The T3 obtained the maximum leaf area index (5.169), followed by the T2 (Table 2). It can be concluded that irrigation with the high concentration biogas slurry better promoted the growth of tomato plants.

**Table 2.** Effects of different biogas slurry proportions and irrigation quantity rates on tomato growth.

| Treatments | Leaf Area/(cm$^2$) | Leaf Area Index | Aboveground Biomass/(g) | Root Weight/(g) | Main Root Length/(cm) | Root/Shoot |
|---|---|---|---|---|---|---|
| CF1 | 6512.39 ± 373.91 g | 3.62 | 163.48 ± 9.76 d | 6.67 ± 0.08 de | 46.92 ± 0.40 b | 0.0408 ± 0.003 |
| CF2 | 7693.52 ± 445.72 f | 4.27 | 179.27 ± 11.16 bcd | 6.83 ± 0.08 cd | 45.61 ± 0.29 cd | 0.0381 ± 0.004 |
| CF3 | 7804.35 ± 369.48 ef | 4.34 | 184.13 ± 10.01 bc | 6.96 ± 0.09 abcd | 44.97 ± 0.61 d | 0.0378 ± 0.003 |
| T1 | 8253.87 ± 428.79 def | 4.59 | 185.14 ± 7.71 bc | 6.85 ± 0.09 bcd | 46.32 ± 0.24 bc | 0.0370 ± 0.002 |
| T2 | 9030.43 ± 266.30 ab | 5.02 | 195.89 ± 9.56 ab | 7.15 ± 0.13 ab | 45.05 ± 0.18 d | 0.0365 ± 0.002 |
| T3 | 9304.99 ± 322.72 a | 5.17 | 211.05 ± 12.18 a | 7.26 ± 0.12 a | 44.75 ± 0.42 d | 0.0344 ± 0.002 |
| T4 | 7063.45 ± 303.56 g | 3.92 | 174.29 ± 8.42 cd | 6.71 ± 0.05 de | 47.03 ± 0.06 b | 0.0385 ± 0.003 |
| T5 | 8596.06 ± 184.64 bcd | 4.78 | 186.79 ± 11.90 bc | 6.93 ± 0.09 bcd | 45.95 ± 0.25 bcd | 0.0371 ± 0.003 |
| T6 | 8902.54 ± 275.95 bc | 4.95 | 188.56 ± 10.64 bc | 7.09 ± 0.13 abc | 45.12 ± 0.49 d | 0.0376 ± 0.002 |
| T7 | 6895.26 ± 387.05 g | 3.83 | 163.43 ± 7.14 d | 6.39 ± 0.11 e | 48.15 ± 0.25 a | 0.0391 ± 0.002 |
| T8 | 8354.75 ± 240.26 cde | 4.64 | 175.20 ± 9.25 cd | 6.64 ± 0.10 de | 46.98 ± 0.44 b | 0.0379 ± 0.02 |
| T9 | 8528.61 ± 152.79 bcd | 4.74 | 183.33 ± 8.88 bc | 6.82 ± 0.10 cd | 45.93 ± 0.37 bcd | 0.0372 ± 0.002 |
| P (biogas slurry) | <0.001 | na | 0.010 | <0.001 | <0.001 | 0.160 |
| P (Irrigation) | <0.001 | na | 0.015 | <0.001 | <0.001 | 0.202 |
| P (biogas slurry × Irrigation) | 0.351 | na | 0.972 | 0.990 | 0.960 | 0.974 |

Note: The numerical value after '±' in the above table represents the standard deviation of different observation indicators, and different lowercase letters indicate significant differences between different treatments (*p* = 0.05).

Plant dry matter accumulation was greatly influenced by the supply and distribution of soil water and nutrients in the root zone. The root-shoot ratio of tomato varied considerably with treatments (Table 2). Different biogas slurry and irrigation levels had a significant effect on the root-shoot ratio, while T3 and CF1 obtained the smallest (0.0344) and the largest (0.0408) the root-shoot ratio, respectively. It indicates that the biogas slurry application was more beneficial to the above-ground biomass accumulation. T3 treatment had the shortest main root length (44.75 cm), but the largest root dry weight, indicating that water and nutrient supply patterns affected the root growth of crops.

*3.2. Tomato Yield and Water Use Efficiency*

With the same level of irrigation, the biogas slurry treatment was more conducive to tomato yield formation compared to the chemical fertilizer treatment (Figure 4). T3 (1:4BS, W3) obtained the highest yield (5382.47 g per plant), which was 15.7% higher than CF3 (chemical fertilizer, W3; 4652.67 g per plant) at the same irrigation level, effectively increasing economic benefits. Under the same fertilizer application, mild deficit irrigation (W2) increased WUE while maintaining higher yield. Tomato yield under the 1:4 biogas slurry concentration and the CF was positively related with irrigation rate. The appropriate irrigation promoted the formation of tomato yield. With the same irrigation level, tomato yield under the biogas slurry ratios showed 1:4 > 1:6 > 1:8 > CF, indicating that the biogas slurry was more effective in increasing tomato yield, and the highest yield was obtained under the biogas slurry ratio of 1:4.

In terms of water use efficiency, the highest WUE (29.7 kg m$^{-3}$) was obtained under T1, followed by T2. T9 had the lowest WUE of 19.8 kg m$^{-3}$. With the same level of fertiliser application, WUE was higher under mild deficit irrigation (W2) and moderate deficit irrigation (W1) than full irrigation (W3). The biogas slurry application improved WUE. The suitable water and fertilization strategies positively influenced WUE.

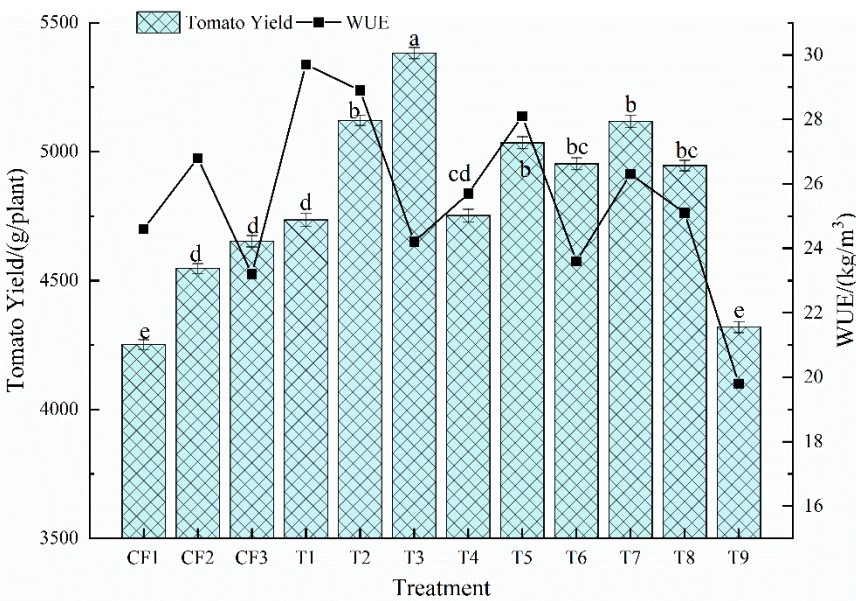

**Figure 4.** Effects of different biogas slurry proportions and irrigation rates on tomato yield and water use efficiency. Different lowercase letters a–e indicate significant differences between treatments.

### 3.3. Tomato Quality

Water and fertilizer regulation patterns affected on various indicators of tomato quality significantly (Table 3). Higher biogas slurry concentration (biogas slurry: water, 1:4) improved tomato quality, such as, soluble sugars of 3.784%, titratable acids of 0.345%, sugar-acid ratio of 10.968, soluble solids of 5.75% and vitamin C of 23.045 mg 100 $g^{-1}$. Under mild irrigation (W2), T2 obtained the optimum fruit quality, except soluble protein and fruit water content, especially the significantly increasing lycopene. T2 increased lycopene of fruits by 20.46% compared to the CF2. Under moderate deficit irrigation (W1), compared to chemical fertilisers, biogas slurry promoted strongly tomato quality, especially sugar-acid ratio and lycopene content. The appropriate biogas slurry concentration effectively improved the nutritional quality of tomatoes. Compared to other fertilization modes, fruit water content was significantly higher under biogas slurry application, and fruit hardness did not reduce. The individual fresh fruit weight under biogas slurry irrigation application increased by more than 7%, and the fruit shape of tomato was optimized (fruit shape index of 0.8 to 0.9).

### 3.4. Comprehensive Nutritional Quality Evaluation

It can be seen that the ranking results of the individual comprehensive nutritional quality models were varied (Table 4). The principal component analysis showed that T3 (1:4BS, W3) was the best treatment for improving fruit quality. However, the other three models showed T2 (1:4BS, W2) ranked the first. Due to the discrepancy of evaluation methods, it was difficult and uncertain to evaluate the comprehensive nutritional quality of tomato. In order to further explore the correlation between the evaluation methods, Pearson correlation analysis was carried out on the evaluation values of the four independent evaluation models, and the correlation coefficient ranged from 0.850 to 0.938 (Table 5). The independent evaluation models had high correlation. Therefore, a comprehensive evaluation model based on the differences and correlations of the independent evaluation models is urgently needed to ensure the scientificity and rationality of the comprehensive nutritional quality evaluation of tomato.

**Table 3.** Effects of different biogas slurry proportion and irrigation quantity on nutritional quality of tomato.

| Treatments | External Quality | | | Taste Quality | | | | Nutrition Quality | | | Storage Quality | |
|---|---|---|---|---|---|---|---|---|---|---|---|---|
| | Weight of Single Fresh Fruit /(g) | Fruit Shape Index | Soluble Sugar/% | Titratable Acid/% | Sugar/Acid Ratio | Soluble Solids/% | Vitamin C /(mg/100 g) | Soluble Protein/(mg/g) | Lycopene/(mg/kg) | Fruit Water Content/% | Fruit Hardness /(kg/cm$^2$) |
| CF1 | 148.62 ± 5.65 d | 0.91 | 1.94 ± 0.14 g | 0.23 ± 0.02 f | 8.53 ± 0.11 | 5.25 ± 0.07 d | 17.92 ± 0.74 g | 0.77 ± 0.05 e | 20.1 ± 0.95 h | 87.4 ± 0.50 e | 7.28 ± 0.22 a |
| CF2 | 150.69 ± 7.81 d | 0.90 | 2.62 ± 0.12 e | 0.25 ± 0.03 def | 10.40 ± 0.55 | 5.35 ± 0.09 cd | 18.65 ± 0.08 fg | 0.89 ± 0.05 d | 22.5 ± 1.15 fg | 88.7 ± 0.35 d | 6.35 ± 0.18 g |
| CF3 | 155.35 ± 5.29 cd | 0.78 | 3.15 ± 0.15 c | 0.30 ± 0.03 abc | 10.48 ± 0.29 | 5.75 ± 0.12 a | 19.97 ± 0.59 de | 0.97 ± 0.04 cd | 21.3 ± 0.95 gh | 89.6 ± 0.26 cd | 6.95 ± 0.17 bcd |
| T1 | 158.21 ± 5.51 bcd | 0.81 | 3.78 ± 0.10 a | 0.35 ± 0.03 a | 10.97 ± 0.32 | 5.43 ± 0.08 bcd | 19.08 ± 057 ef | 0.92 ± 0.03 d | 27.7 ± 1.20 bc | 87.4 ± 0.75 e | 6.53 ± 0.08 fg |
| T2 | 171.12 ± 7.51 a | 0.91 | 3.42 ± 0.10 b | 0.32 ± 0.01 ab | 10.72 ± 0.13 | 5.75 ± 0.13 bc | 22.46 ± 0.57 ab | 1.11 ± 0.04 a | 29.5 ± 1.15 ab | 89.7 ± 0.62 bcd | 7.05 ± 0.09 abc |
| T3 | 173.95 ± 7.15 a | 0.89 | 2.79 ± 0.15 de | 0.27 ± 0.04 cdef | 10.74 ± 0.81 | 5.39 ± 0.14 cd | 23.05 ± 0.52 a | 1.125 ± 0.05 a | 26.2 ± 2.54 bc | 90.7 ± 0.6 b | 6.45 ± 0.15 fg |
| T4 | 165.23 ± 5.45 abc | 0.85 | 3.10 ± 0.12 c | 0.31 ± 0.02 abc | 10.04 ± 0.20 | 5.50 ± 0.11 bc | 21.54 ± 0.68 bc | 1.06 ± 0.09 abc | 30.2 ± 0.95 a | 89.5 ± 0.82 cd | 6.65 ± 0.08 ef |
| T5 | 166.54 ± 3.98 ab | 0.84 | 2.58 ± 0.15 e | 0.27 ± 0.03 cdef | 9.62 ± 0.45 | 5.35 ± 0.10 cd | 18.57 ± 0.68 fg | 0.89 ± 0.05 d | 24.7 ± 1.51 def | 90.4 ± 0.56 bc | 6.53 ± 0.11 fg |
| T6 | 165.84 ± 4.32 ab | 0.86 | 2.98 ± 0.18 cd | 0.29 ± 0.02 bcd | 10.19 ± 0.15 | 5.48 ± 0.12 bc | 20.83 ± 0.50 cd | 1.08 ± 0.08 ab | 24.3 ± 0.92 def | 90.2 ± 0.56 bc | 6.85 ± 0.13 de |
| T7 | 168.63 ± 4.59 ab | 0.85 | 3.028 ± 0.13 cd | 0.27 ± 0.02 cdef | 11.19 ± 0.33 | 5.31 ± 0.11 cd | 18.76 ± 0.57 fg | 1.09 ± 0.07 ab | 25.8 ± 1.15 cde | 88.9 ± 0.56 d | 7.14 ± 0.11 ab |
| T8 | 165.28 ± 3.43 abc | 0.82 | 2.31 ± 0.08 f | 0.24 ± 0.02 ef | 9.47 ± 0.37 | 5.57 ± 0.12 ab | 18.38 ± 0.55 fg | 1.13 ± 0.04 a | 26.6 ± 1.21 cd | 91.8 ± 0.75 a | 6.3 ± 0.14 g |
| T9 | 149.08 ± 3.85 d | 0.91 | 2.86 ± 0.14 d | 0.29 ± 0.02 bcde | 9.99 ± 0.28 | 5.45 ± 0.12 bcd | 20.01 ± 0.38 de | 0.99 ± 0.07 bcd | 23.7 ± 1.15 ef | 89.6 ± 0.56 bcd | 6.79 ± 0.11 de |

Note: Different lowercase letters a–h indicate significant differences between treatments.

**Table 4.** Result of different evaluation models on comprehensive nutrition quality of tomato.

| Treatments | Principal Component Analysis | | Grey Correlation Method | | Membership Function Analysis Method | | The TOPSIS Model Based on Combination Weighting | | Overall Difference Combination Evaluation Model | |
|---|---|---|---|---|---|---|---|---|---|---|
| | Evaluation Value | Ranking | Evaluation Value | Ranking | Evaluation Value | Ranking | Evaluation Value | Ranking | Evaluation Value | Ranking |
| CF1 | 2.1911 | 12 | 0.6127 | 12 | 0.1531 | 12 | 0.1461 | 12 | −1.7634 | 12 |
| CF2 | 2.2494 | 11 | 0.6422 | 11 | 0.2866 | 11 | 0.3032 | 11 | −1.1372 | 11 |
| CF3 | 2.3036 | 9 | 0.7147 | 8 | 0.5063 | 7 | 0.4964 | 7 | −0.2128 | 7 |
| T1 | 2.3338 | 8 | 0.7691 | 4 | 0.4813 | 6 | 0.6183 | 3 | 0.1583 | 6 |
| T2 | 2.5106 | 2 | 0.8943 | 1 | 0.8497 | 1 | 0.8467 | 1 | 1.7971 | 1 |
| T3 | 2.5190 | 1 | 0.8126 | 2 | 0.6654 | 2 | 0.6286 | 4 | 1.0076 | 2 |
| T4 | 2.4586 | 3 | 0.7952 | 3 | 0.6384 | 3 | 0.7002 | 2 | 0.8649 | 3 |
| T5 | 2.3759 | 7 | 0.6601 | 10 | 0.3764 | 10 | 0.3305 | 10 | −0.6196 | 10 |
| T6 | 2.4117 | 4 | 0.7485 | 6 | 0.5904 | 4 | 0.5743 | 6 | 0.3715 | 4 |
| T7 | 2.3944 | 5 | 0.7533 | 5 | 0.5387 | 5 | 0.5087 | 5 | 0.1873 | 5 |
| T8 | 2.3962 | 6 | 0.7073 | 7 | 0.4578 | 8 | 0.3822 | 9 | −0.2367 | 8 |
| T9 | 2.2829 | 10 | 0.6982 | 9 | 0.4611 | 9 | 0.4677 | 8 | −0.4160 | 9 |

**Table 5.** Pearson correlation coefficient of evaluation value of each evaluation model.

| Pearson Correlation Coefficient | Principal Component Analysis | Grey Correlation Method | Membership Function Analysis Method | The TOPSIS Model Based on Combination Weighting | Mean Value |
|---|---|---|---|---|---|
| Principal component analysis | | 0.860 | 0.891 | 0.800 | 0.850 |
| Grey correlation method | 0.860 | | 0.965 | 0.970 | 0.932 |
| Membership function analysis method | 0.891 | 0.965 | | 0.958 | 0.938 |
| The TOPSIS model based on combination weighting | 0.800 | 0.970 | 0.958 | | 0.909 |
| Overall difference combination evaluation model | 0.918 | 0.984 | 0.989 | 0.967 | 0.965 |

### 3.5. Nutritional Quality Evaluation Based on an Overall Difference Combination Evaluation Model

#### 3.5.1. Statistical Test for the Nutritional Quality Combination Evaluation Model

It is important to do the consistency test for ensuring the scientificity and rationality of the combined evaluation model before performing [32]. The Kendall consistency coefficient test is used in this study, while results showed that the coefficient of consistency test (T) was 0.927, the test statistic ($\chi^2$) was 40.788, and $\chi^2_{0.05}(11) = 24.725$. The compatibility of four independent evaluation models was found. The requirements of consistency test were satisfied. Hence, the combined evaluation model can be performed directly.

#### 3.5.2. Comprehensive Quality Evaluation Based on the Overall Difference Combination Evaluation Model

The combined evaluation model was used to estimate the evaluation values of the four independent evaluation models. The quality indices of fruits were ranked. The larger the evaluation value, the higher ranking of the evaluation object. T2 (1:4BS, W2) was the first, and CF1 (CF, W1) ranked the last (Table 4). To verify the correlation between quality evaluation models, the Pearson two-by-two correlation analysis was performed. Results showed that the mean value of the correlation coefficient reached 0.965 (Table 5). The combined evaluation model had a good correlation with each independent comprehensive evaluation model (0.918–0.989). It further demonstrated the scientificity and rationality of the combined evaluation model. This indicated that the results of the combined evaluation model were reliable.

## 4. Discussion

As an after-product of biogas engineering, biogas slurry is a quick-acting organic fertilizer with sufficient reserves. A reasonable biogas slurry application pattern is potential to increase crop yields and improve soil environment in the cultivated area [33]. High and stable crop yield is the main objective of agricultural production, and how to improve yield

has always been a vital research area. According to the previous studies and farming experience, high water and fertilizer supply is the most direct way to improve high crop yield. However, excessive water and fertilizer application not only causes serious agricultural non-point pollution [34], but also has a certain inhibitory effect on crop growth and water and fertilizer use efficiency. Additionally, compared with chemical fertilizer treatments, biogas slurry concentration of 1:4 effectively promoted the formation of tomato leaf area and root growth as well as optimized the root-shoot ratio, so as to improve tomato yield and water use efficiency. The reasons may be that biogas slurry had the good solubility [35], which is conducive to the rapid absorption of nutrients required for crop growth. The appropriate leaf area improves the transformation of photoassimilate. Sufficient water and nutrient supply boosted root absorption, avoiding the root overgrowth and consumption of photosynthetic products due to lack of water and nutrient. Meanwhile, hole irrigation with biogas slurry applied in this study enhanced the horizontal infiltration in the root zone soil, decreased the evaporation, reduced the risk of a dense layer formation on the infiltration surface resulting from organic suspended particles in the biogas slurry, promoted the migration of biogas fertilizer in the soil [36], and effectively improved the water and fertilizer absorption of the root. In this study, it was found that higher yield and water use efficiency were obtained under high biogas slurry concentration and mild water deficit. The reason may be the fact that high biogas slurry concentration with the high viscosity, infiltrating with irrigation water, adhered to the surface soil (0–20 cm) promoting the water and nutrient absorption of tomato roots. The biogas slurry has the characteristics of high water and low fertilizer, and biogas slurry at low concentrations is easy to deep leak inhibiting soil nutrients preservation and soil aggregate structure formation [15]. Moderate water deficit can effectively regulate the tomato plants growth and nutrient distribution.

Crop quality is influenced by numerous factors, among which crop varieties based on different genetic traits affect quality strongly [37]. However, a large amount of studies have shown that ameliorating environmental factors and optimizing water and fertilizer management for crop growth are essential ways to control fruits quality artificially [38,39]. High quality crop fruit is produced from the synthesis and accumulation of effectively photosynthetic assimilation products as well as plant nutrient growth metabolism. When measures are taken to facilitate the assimilates transfer to the fruits and alter the metabolic pathways, promoting fruit quality can be achieved [40]. Different water and fertilizer regulation patterns influenced tomato quality (Table 2). The results showed that biogas slurry increased the individual fresh fruit weight of tomatoes, compared to chemical fertilizers. The effect of biogas slurry on individual fresh fruit weight was significant at high concentrations, and also biogas slurry effectively improved the appearance of tomato fruits. The main reason is the fact that biogas slurry, a fast-acting liquid organic fertilizer, was applied to the irrigation holes facilitating the rapid water and nutrients transport to the soil in the root zone, enhancing water and fertilizer uptake by the tomato plant, and improving the conversion of nutrients to the fruit. Meanwhile, mild deficit irrigation (W2), all quality indicators of tomato fruits obtained optimum values under T2. It was because the suitable deficit irrigation regulation increased the photosynthesis of tomato plants, resulting in more assimilates to the fruit. The enzymes in the biogas slurry further promoted the assimilation of water and nutrients in the xylem of tomato. It is consistent with the findings of Zheng et al. [14] who proposed biogas slurry irrigation significantly enhanced nutritional quality of tomatoes.

Interestingly, the nutritional quality of tomato fruits is determined by the interaction of multiple indicators. Different people have various quality requirement of fruits. Tomato producers focus on external and storage quality of fruits, conversely, taste quality and nutrition quality are vital for consumers. Thus, the objective, scientific and rational evaluation of tomato fruits and the selection of appropriate evaluation indicators and models are of great importance for tomato water and fertilizer regulation. At present, many scholars have adopted a comprehensive evaluation method based on multi-attribute decision making for the evaluation of the comprehensive nutritional quality of tomatoes. However, the

estimation results from different methods often varied greatly, due to the differences of evaluation methods and thinking logic, mainly in the selection of evaluation indicators and the determination of attribute weights. Similarly, the evaluation results in this study of four comprehensive evaluation methods varied considerably, mainly due to the different focuses and perspectives of each method in the analysis of the evaluation objects. In order to avoid contradictions in the guidance of different evaluation methods for production practices, this study firstly analyzed the results of Kendall consistency test and Pearson correlation of the independent evaluation methods. There was a high correlation among four independent evaluation models (mean value higher than 0.85), indicating that a combined evaluation model could be established to ensure the scientificity and rationality of the evaluation models. Results showed that the quality of tomatoes fruits was evaluated based on the combined evaluation model which has good correlation with the four independent comprehensive evaluation models, enables a better integration of models and makes the evaluation of fruits quality more scientific and reasonable.

### 5. Conclusions

(1) Mild deficit irrigation (W2) and full irrigation (W3) under the integrated water/biogas slurry irrigation technology improved tomato leaf area and root formation, compared to the chemical fertilizer treatments. The T3 (1:4BS, W3) obtained the greatest leaf area and root dry mass.

(2) Under the same irrigation rate, the biogas slurry enhanced tomato yield and effectively increased water use efficiency, compared to the chemical fertilizer treatment. The highest tomato yield was 5382.47 g per plant under T3 (1:4BS, W3), followed by the T2 (1:4BS, W2). The water use efficiency of the biogas slurry application was generally higher than that of the chemical fertilizer treatment, and a maximum water use efficiency of 29.7 kg/m$^3$ was obtained under T1 (1:4BS, W1).

(3) There were significant differences in the ranking obtained from four independent evaluation methods. Results from the Kendall correlation test showed that the independent evaluation methods have good compatibility. The Pearson correlation coefficient among the independent evaluation models ranged from 0.850 to 0.938 with high consistency. The average Pearson correlation coefficient between the combined evaluation model and the each of four independent quality evaluation models reached 0.965. The combined evaluation model not only reflected the evaluation results scientifically and reasonably, but also alleviated the discrepancy of the independent comprehensive evaluation models. The evaluation results showed that T2 (1:4BS, W2) was strongly recommended in this study.

**Author Contributions:** J.Z.: investigation, conceptualization, visualization, methodology. X.Q.: investigation, data curation, validation, conceptualization, writing—original draft, writing—review and editing. C.S. and S.Y.: investigation, data curation, formal analysis. Y.W.: investigation, data curation, software. All authors have read and agreed to the published version of the manuscript.

**Funding:** The National Natural Science Foundation of China (51969012) and Red Willow First-class Discipline Project of Lanzhou University of Technology (0807J1), and Industry Supporting and Guiding Project of Gansu Higher Education Institutions (2021CYZC-27, 2021CYZC-33).

**Institutional Review Board Statement:** Not applicable.

**Informed Consent Statement:** Not applicable.

**Conflicts of Interest:** The authors declare that they have no conflict of interest.

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
