# Peer review of "Tomato Comprehensive Quality Evaluation and Irrigation Mode Optimization with Biogas Slurry Based on the Combined Evaluation Model"

_agronomy, doi:10.3390/agronomy12061391_

Round 1
Reviewer 1 Report
Review
The article deals with the effect of digestate on tomato yield and quality. The presented evaluation model is comprehensive and brings new insight into the evaluation issue. It takes into account a wide range of parameters instead of evaluating each part individually. From this point of view, the paper is very well written. The quality is reduced by some minor errors which need to be corrected.
L 248: typo – elven.
L 256: Statistical analysis and analysis – typo I guess.
The entire section 2.6 needs to be completed with information about the SW manufacturer. The editor would call for this in later parts of the review process. Information on statistical analysis and post-hoc tests is missing.
The sentence beginning on line 266 makes no sense. The P-value is high for a significant effect.
Regarding the tables, please reduce the number of digits. It's useless to have such precision. For calculations from equations, it is of course best to have high precision. But for standalone tables, it is unnecessary and useless (I give tables 2 and 3 as an example).
The description of Table 2 has been omitted.
Go through the whole text again and check for errors (see L320, -1 and so on).
Author Response
General Response:
According to the suggestions of the editor and referees’ comments, we have made a substantial revision to the original manuscript such that a clear description on the research is displayed in the revised version. We deeply appreciate the time and effort you have spent in reviewing our manuscript. The detailed responses to the comments of the referees are as follows:
Point 1: L 248: typo – elven.
Response 1: Thank you for the kindly and careful comment and recommendation. According to the suggestion, the word "elven" has been revised to "eleven" in line 259 of the revised manuscript.
Point 2: Statistical analysis and analysis – typo I guess.
Response 2: Thank you for the carefully reviewing of the manuscript. We are very sorry for this negligence! According to the suggestion, the heading "Statistical analysis and analysis" has been revised to "Statistical analysis" in line 267 of the revised manuscript.
Point 3: The entire section 2.6 needs to be completed with information about the SW manufacturer. The editor would call for this in later parts of the review process. Information on statistical analysis and post-hoc tests is missing.
Response 3: Thank you for the kindly and careful comment and recommendation. According to the suggestion, the related sentences “The entropy weight method weight, grey correlation analysis, membership function analysis method, TOPSIS method and Kendall consistency test were performed by Excel 2019. Principal component analysis, variance analysis and Pearson correlation analysis were performed by SPSS24.0. MATLAB6.5 software was used to perform the overall difference combination evaluation model.” have been revised into “The entropy weight method weight, grey correlation analysis, membership function analysis method, TOPSIS method and Kendall consistency test were performed by Excel 2019 (Office2019. Microsoft Corp, USA). Principal component analysis, variance analysis and Pearson correlation analysis were performed by SPSS24.0 (OriginLab Corporation, Northampton, MA, USA). MATLAB6.5 software (MathWorks, USA) was used to perform the overall difference combination evaluation model. One–way ANOVA and Duncan’s multiple comparisons were used to determine significantdifferences between the treatments.” in lines 268-275 of the revised manuscript.
Point 4: The sentence beginning on line 266 makes no sense. The P-value is high for a significant effect.
Response 4: Thank you for the kindly and careful comment and recommendation. According to the suggestion, the related sentences “Different irrigation levels and biogas slurry application had significant effects on tomato leaf area, above-ground dry matter, root dry weight, main root length and root shoot ratio (Table 2).” have been revised into “Reasonable irrigation levels and biogas slurry application can effectively promote the formation tomato leaf area, above-ground dry matter, root dry weight, main root length and root shoot ratio (Table 2).” in lines 278-280 of the revised manuscript.
Point 5: Regarding the tables, please reduce the number of digits. It's useless to have such precision. For calculations from equations, it is of course best to have high precision. But for standalone tables, it is unnecessary and useless (I give tables 2 and 3 as an example).
Response 5: Thank you for the kindly and careful comment and recommendation. According to the suggestion, we have adjusted the precision in the table to two digits.
Point 6: The description of Table 2 has been omitted.
Response 6: Thank you for the kindly and careful comment and recommendation. We are very sorry for this negligence! According to the suggestion, the description of Table 2 has been added in line 295 of the revised manuscript.
Point 7: Go through the whole text again and check for errors (see L320, -1 and so on).
Response 7: Thank you for the carefully reviewing of the manuscript. We are very sorry for this negligence! According to the suggestion, we have read the full text, checked and corrected errors in the text.

Reviewer 2 Report
The manuscript adopted a variety of data analysis methods to evaluate and analyze the comprehensive quality of tomatoes under integrated different water/biogas slurry irrigation conditions, and obtained some meaningful results. The research has a lot of research value and practical significance and the manuscript was well arranged. However, there were still some shortcomings:
(1) In lines 79-90, the author explained the shortcomings of the existing agricultural production evaluation methods, but lacked the detailed explanation on applicability of the overall difference combination evaluation model used in this paper, please add it.
(2) In Section "Experimental design", the description of the biogas slurry application method is insufficient in the experimental design, and revise it.
(3) In lines 247-253, the authors described the comprehensive quality evaluation index system for tomatoes, but not referred to the methods. Therefore, in line 246 there should be the “tomato comprehensive quality evaluation index system”, not “evaluation method”, please revise it.
(4) Figure 1 and Figure 4 are not clear enough, please revise and make them clearer.
(5) In lines 303-305, the author proposes “Under the same fertilizer application, mild deficit irrigation (W2) increased WUE while maintaining higher yield. Tomato yield under the 1:4 biogas slurry concentration and the CF was positively related with irrigation rate.” But it can be seen from Figure 4 that T1 treatment achieved the highest WUE, please explain what is the meaning of “Under the same fertilizer application”?
(6) In lines 325-326, the author proposes “Compared to other fertilization modes, fruit water content was significantly higher under biogas slurry application, and fruit hardness did not reduce.” Please explain the meaning of “other fertilization modes” in the manuscript.
(7) In lines 336-339, high correlations between independent models is the key index to propose a comprehensive evaluation model, but there is no specific value or standard to define a high correlation, please add the explanation or references.
(8) In lines 398-401, the authors put forward “The reason may be the fact that high biogas slurry concentration with the high viscosity, infiltrating with irrigation water, adhered to the surface soil (0-20cm) promoting the water and nutrient absorption of tomato roots.” It seems that the higher the biogas slurry concentration, the better the water and nutrients absorption by the crop roots. However, the existing research results show that this is not the case, while there is a certain range of biogas slurry concentration applications, please revise the relative contents.
Author Response
General Response:
According to the suggestions of the editor and referees’ comments, we have made a substantial revision to the original manuscript such that a clear description on the research is displayed in the revised version. We deeply appreciate the time and effort you have spent in reviewing our manuscript. The detailed responses to the comments of the referees are as follows:
Point 1: In lines 79-90, the author explained the shortcomings of the existing agricultural production evaluation methods, but lacked the detailed explanation on applicability of the overall difference combination evaluation model used in this paper, please add it.
Response 1: Thank you for the kindly and careful comment and recommendation. According to the suggestion, the related statements “The study shows that the overall difference combination evaluation model can overcome the inconsistent findings of multiple single evaluation methods and make the results more objective and accurate.[29]” have been added in lines 90-92 of the revised manuscript.
And the cited contents are:
[29] Yan Fulai, Zhang Fucang, Fan Xingke, et al. Optimal irrigation and nitrogen application amounts for spring maize based on evaluation model in study soil area in Ningxia[J]. Transactions of the Chinese Society for Agricultural Machinery, 2020, 51(09): 258-265.
Point 2: In Section "Experimental design", the description of the biogas slurry application method is insufficient in the experimental design, and revise it.
Response 2: Thank you for your very valuable comments and suggestions. According to the suggestion, the related statements on the specific operation of biogas slurry treatment “‘It should be noted that the biogas slurry was placed in a plastic bucket and left open to settle for two months before the experiment, when used the upper clear liquid was taken and the larger suspended particles were filtered out using 4 layers gauze of 32 mesh. Meanwhile, the relevant physical and chemical properties of the biogas slurry were measured every seven days during the test to ensure the basic stability of the physical and chemical properties of the biogas slurry used during the test period.” have been added in lines 139-144 of the revised manuscript.
Point 3: In lines 247-253, the authors described the comprehensive quality evaluation index system for tomatoes, but not referred to the methods. Therefore, in line 246 there should be the “tomato comprehensive quality evaluation index system”, not “evaluation method”, please revise it.
Response 3: Thank you for your very valuable comments and suggestions. We are very sorry for this negligence! According to the suggestion, the related heading “evaluation method” have been revised into ” Tomato comprehensive quality evaluation index system” in line 257 of the revised manuscript.
Point 4: Figure 1 and Figure 4 are not clear enough, please revise and make them clearer.
Response 4: Thank you for the carefully reviewing of the manuscript. We are very sorry for this negligence! According to the suggestion, in order to make the figure clearer, the figure has been revised , shown as follow:
Figure 1 The location of experiment site.
Figure 4. Effects of different biogas slurry proportions and irrigation rates on tomato yield and water use efficiency.
Point 5: In lines 303-305, the author proposes “Under the same fertilizer application, mild deficit irrigation (W2) increased WUE while maintaining higher yield. Tomato yield under the 1:4 biogas slurry concentration and the CF was positively related with irrigation rate.” But it can be seen from Figure 4 that T1 treatment achieved the highest WUE, please explain what is the meaning of “Under the same fertilizer application”?
Response 5: Thank you for the carefully reviewing of the manuscript. In this study, the same fertilization method refers to the application of the same concentration of biogas slurry or the same amount of chemical fertilizer. Among them, the biogas slurry concentrations were 1:4, 1:6 and 1:8, respectively.Fertiliser application was based on the amount applied by local farmers.
Point 6: In lines 325-326, the author proposes “Compared to other fertilization modes, fruit water content was significantly higher under biogas slurry application, and fruit hardness did not reduce.” Please explain the meaning of “other fertilization modes” in the manuscript.
Response 6: Thank you for the carefully reviewing of the manuscript. The meaning of “other fertilization modes” is the chemical fertilizer application under different irrigation rates.
Point 7: In lines 336-339, high correlations between independent models is the key index to propose a comprehensive evaluation model, but there is no specific value or standard to define a high correlation, please add the explanation or references.
Response 7: Thank you for the carefully reviewing of the manuscript. In this study, in order to improve the accuracy and applicability of the overall difference portfolio evaluation model, Pearson correlation analysis was used for ex ante testing, and the results showed that the number of prior relationships ranged from 0.850 to 0.938, and therefore the independent evaluation models had a high degree of consistency.
And the cited contents are:
[25]Hong Xia. Combination evaluation model of tomato yield quality environment effect and ITS response to water and fertilizer in greenhouse[D]. Northwest A&F University, 2018.
[29] Yan Fulai, Zhang Fucang, Fan Xingke, et al. Optimal irrigation and nitrogen application amounts for spring maize based on evaluation model in study soil area in Ningxia[J]. Transactions of the Chinese Society for Agricultural Machinery, 2020, 51(09): 258-265.
Point 8: In lines 398-401, the authors put forward “The reason may be the fact that high biogas slurry concentration with the high viscosity, infiltrating with irrigation water, adhered to the surface soil (0-20cm) promoting the water and nutrient absorption of tomato roots.” It seems that the higher the biogas slurry concentration, the better the water and nutrients absorption by the crop roots. However, the existing research results show that this is not the case, while there is a certain range of biogas slurry concentration applications, please revise the relative contents.
Response 8: Thank you for the very valuable comment and recommendation. Following the reviewer's suggestion, we have revised this unclear description carefully and added the lost information. In this study, the high concentration of biogas slurry was 1:4, which is relative to the 1:6 and 1:8 concentrations. According to the previous research by our group, the optimal soil environment can be obtained when the concentration of biogas slurry is 1:4. At the same time, when the biogas slurry infiltrated the soil in the root zone, it had the greatest impact on the soil layer of 0-20 cm, and the appropriate biogas slurry concentration (1:4) was just conducive to the formation of soil aggregate structure and the improvement of microbial activity in the soil.
And the cited contents are:
Jian Zheng, Pingan Zhang, Ningbo Cui, et al. Experimental Study on the Optimal Threshold of Water and Nitrogen Quantities for Tomato Growth under the Irrigation with Biogas Slurry in Greenhouse[J]. Applied Engineering in Agriculture. 2021, 37(2): 333-342.
Zheng J., Ma J., Feng Z. J, et al. Effects of biogas slurry irrigation on tomato physiological and ecological indexes, yield and quality as well as soil environment[J]. Applied Ecology and Environmental Research 2020, 18(1):1013-1029.
Zheng Jian, Li Xinyi, Zhang Yanning, et al. Effects of Digestate Application on Tomato Growth, Yield, Quality, and Soil Nitrogen Content via Integrated Hole Irrigation[J]. Journal of Biobased Materials and Bioenergy, 2019,13(5): 620–634.
Zheng Jian, Pan Zhanpeng, Ma Jing, et al. Animal Based Biogas Digestate Application Frequency Effects on Growth and Water-Nitrogen use Efficiency in Tomato[J]. International Journal of Agriculture & Biology, 2019, 22(4): 748-756.
